Definition of sampling units begets conclusions in ecology: the case of habitats for plant communities

Mörsdorf Martin A. 1 2 3 mam28@hi.is
Ravolainen Virve T. 2 4
Støvern Leif Einar 5
Yoccoz Nigel G. 2
Jónsdóttir Ingibjörg Svala 1 3
Bråthen Kari Anne 2
1 Department of Life and Environmental Sciences, University of Iceland , Reykjavík , Iceland
2 Department of Arctic and Marine Biology, The Arctic University of Norway , Tromsø , Norway
3 Department of Arctic Biology, University Centre in Svalbard UNIS , Longyearbyen , Norway
4 Norwegian Polar Institute , Tromsø , Norway
5 Norwegian Institute for Forest and Landscape Research , Tromsø , Norway
Gandini Patricia
Electronic publication date: 2015 Mar 5
Publication date: 2015
Volume: 3
Electronic Location ID: e815
Received 2014 Oct 27; Accepted 2015 Feb 13
Copyright: © 2015 Mörsdorf et al.
Copyright year: 2015
Copyright holder: Mörsdorf et al.
License: This is an open access article distributed under the terms of the Creative Commons Attribution License, which permits unrestricted use, distribution, reproduction and adaptation in any medium and for any purpose provided that it is properly attributed. For attribution, the original author(s), title, publication source (PeerJ) and either DOI or URL of the article must be cited.
License URL: https://creativecommons.org/licenses/by/4.0/

Keywords: Sampling design, Expert knowledge, Formal rules, Sampling frame, Snowbed habitat, Mesic habitat

Funding: Norwegian Environment Agency Icelandic Research Fund Climate-Ecological Observatory of Arctic Tundra (COAT) This work was initiated and received financial support from the Norwegian Environment Agency program of monitoring in mountainous areas. We also acknowledge financial support from the Icelandic Research Fund and from the Climate-Ecological Observatory of Arctic Tundra (COAT). The funders had no role in study design, data collection and analysis, decision to publish, or preparation of the manuscript.

==============================
In ecology, expert knowledge on habitat characteristics is often used to define sampling units such as study sites. Ecologists are especially prone to such approaches when prior sampling frames are not accessible. Here we ask to what extent can different approaches to the definition of sampling units influence the conclusions that are drawn from an ecological study? We do this by comparing a formal versus a subjective definition of sampling units within a study design which is based on well-articulated objectives and proper methodology. Both approaches are applied to tundra plant communities in mesic and snowbed habitats. For the formal approach, sampling units were first defined for each habitat in concave terrain of suitable slope using GIS. In the field, these units were only accepted as the targeted habitats if additional criteria for vegetation cover were fulfilled. For the subjective approach, sampling units were defined visually in the field, based on typical plant communities of mesic and snowbed habitats. For each approach, we collected information about plant community characteristics within a total of 11 mesic and seven snowbed units distributed between two herding districts of contrasting reindeer density. Results from the two approaches differed significantly in several plant community characteristics in both mesic and snowbed habitats. Furthermore, differences between the two approaches were not consistent because their magnitude and direction differed both between the two habitats and the two reindeer herding districts. Consequently, we could draw different conclusions on how plant diversity and relative abundance of functional groups are differentiated between the two habitats depending on the approach used. We therefore challenge ecologists to formalize the expert knowledge applied to define sampling units through a set of well-articulated rules, rather than applying it subjectively. We see this as instrumental for progress in ecology as only rules based on expert knowledge are transparent and lead to results reproducible by other ecologists.

Introduction

Sampling in ecology can be challenging. Ecological systems are characterized by complexity (Loehle, 2004) about which there is a paucity of information (Carpenter, 2002). Hence, ecological sampling is often accompanied by unknown characteristics that may unintentionally cause estimates to be dependent on the sampling designs, even to the extent that they “beget conclusions”, as was shown for the impact of the Exxon Valdez oil spill (Peterson et al., 2001; Peterson et al., 2002). The bases for achieving unbiased estimates are study- or sampling designs that include well-articulated objectives along with proper methodology (Olsen et al., 1999; Yoccoz, Nichols & Boulinier, 2001; Albert et al., 2010). In addition, sampling designs need to be transparent, enabling others to repeat the study. Accordingly, ecologists have been encouraged to use formal approaches (Legendre et al., 2002; Edwards et al., 2005; Edwards et al., 2006; Albert et al., 2010). However, whilst sources of bias and a call for formal rules in sampling designs have received attention, the seemingly simple task of defining a sampling unit, such as a study site, also merits thorough consideration, especially in community ecology. Indeed, the definition of sampling units is often a task that demands expert knowledge. Expert knowledge can be applied in such a way that sampling units are formally defined but in ecology, expert knowledge implies often a subjective definition of sampling units before data collection is initiated (Whittaker, Levin & Root, 1973; Kenkel, Juhász-Nagy & Podani, 1989; Franklin, Noon & George, 2002; Loehle, 2004; McBride & Burgman, 2012).

In situations where sampling units are not clearly defined, the availability of relevant sampling units is not known before entering the field, i.e., there is no well-defined sampling frame and in its vacancy, a subjective definition of sampling units is applied in order to guide sampling to ecological units that are determined to be suitable in the field. In principle, the selection of any subjectively defined sampling unit can never be sufficiently articulated as to enable other researchers to repeat the study, or to allow generalizations of results to a specific target population (in a statistical sense) (Olsen et al., 1999; Schreuder, Gregoire & Weyer, 2001). Moreover, in phytosociological studies it has been documented that individual preferences in selecting sampling units that were defined subjectively can lead to biased estimates (Chytrý, 2001; Botta-Dukát et al., 2007; Hédl, 2007). The criticism of using a subjective definition of sampling units is both theoretically and empirically based, but it may merely reflect a study-specific bias between subjective and more formal approaches. Therefore, in this study we want to compare a subjective versus a formal definition of sampling units in the same study system in order to assess whether subjective definition merely introduces bias, overstating findings, or if the way of defining sampling units even begets conclusions.

Studies that have compared formal versus subjective sampling have investigated sampling units based on existing geographical data (e.g., Edwards et al., 2006; Hédl, 2007; Michalcová et al., 2011). A formal a priori definition of sampling unit criteria could therefore be done before sampling was initiated. However, ecologists might often not have access to such data which is especially the case when the spatial extent of sampling units is smaller than the spatial resolution of previously existing geographical data (Roleček et al., 2007). As a way of making the definition of units more transparent in such situations, a few studies used formal criteria for suitable sampling units that are defined a priori to the field sampling (e.g., Ravolainen et al., 2010).

Habitats are perhaps some of the most difficult sampling units to define (Whittaker, Levin & Root, 1973; Franklin, Noon & George, 2002), but are central to many conservation programs such as the “European council directive on the conservation of natural habitats and of wild fauna and flora” (FFH) (The Council of the European Communities, 1992) or the International Union for Conservation of Nature (IUCN) Red List of Threatened Species (IUCN, 2013). Despite the acknowledged importance of habitats, definitions differ greatly among conservation programs worldwide. Whereas some conservation initiatives rely on formal definitions of habitat criteria (Jeffers, 1998; Jongman et al., 2006), others rely on a subjective definition of habitats in the field (Jennings et al., 2009). In this paper, we focus on habitats and address the question of whether subjective or formal definitions of sampling units lead to different estimates of habitat properties. We therefore compared a formal approach, where the final selection of these habitats involved an a priori definition of sampling units, to an approach involving only a subjective definition (sensu Gilbert, 1987).

For both approaches we aimed at two habitats typical for tundra. These habitats are characterized by their difference in growing conditions and are found in sloping, concave terrain. Here, slopes of intermediate steepness provide intermediate moisture conditions (mesic habitats) and gently inclined slopes have wetter conditions combined with a long lasting snow cover (snowbed habitats) (Fremstad, 1997). For the formal approach of defining sampling units, we used explicit criteria of the aforementioned habitat terrain and a terrain model in order to extract a list of potential sampling units. Because we expected that some of these would not be suitable for sampling (e.g., because of boulder fields), we pre-defined additional habitat criteria to be applied in the field. For the subjective approach of defining sampling units, habitats were solely subjectively selected in the field. Both approaches were applied within the same sampling design that ensured balanced sampling with respect to major ecological gradients.

The research questions, i.e., what are the plant community characteristics that describe mesic and snowbed habitats, and the measurement of plant community characteristics, were the same in both approaches. For all sampling units, estimates of standing crop of the most abundant plant species and plant functional groups were assessed as well as within plant community diversity. Finally, to evaluate whether different approaches to defining sampling units lead to different estimates of habitat properties, we tested the effect of using formal versus subjective definition of sampling units on the estimates of these plant community characteristics.

Materials and Methods

Ecosystem characteristics

The field sampling for the current study was conducted during peak growing season between 20th and 30th of July 2011 on Varanger Peninsula, the north-eastern part of Finnmark County in northern Norway (Fig. 1A). The Varanger Peninsula is delineated by the Barents Sea towards the north and birch forests towards the south. Sandstone, sandstone intermingled with schist, and sandstone intermingled with schist and calcareous bedrock are among the most common geological parental materials (The Geological survey of Norway; www.ngu.no). The topography is characterized by a mixture of plateaus and gently sloping hills (maximum height of approximately 500 m) that are intersected by river valleys. The plateaus build a border with steep slopes towards the Barents Sea. During the growing season (July to August) average (monthly) precipitation is 47.7 mm (range 38–55 mm) and temperature is 8.7 °C (range 6.2–10.5 °C) (30 year averages from 1960 to 1990, Norwegian Meteorological Institute, www.met.no).

Figure 1 The figure represents the hierarchical nestedness of the sampling design.

(A) The figure shows the geographical location of the sampling region (Varanger Peninsula, northern Norway) and nestedness of the sampling design. The shades of gray delimit the districts of contrasting reindeer density. Open squares show the raster of 2 × 2 km landscape areas where major roads, power lines, glaciers and large water bodies have been omitted. Black squares correspond to landscape areas that adhered to all other delimitations in our design (see Materials and Methods section for details). (B) One landscape area contained up to two study areas (dashed line) which inherited a pair of formally (GPS) and subjectively (eye) defined sampling units. (C) Each sampling unit contained both a mesic and a snowbed habitat. The recording of vegetation characteristics within each habitat was conducted along transects (dashed lines within habitats).

We conducted our study in the low alpine zone. The vegetation of the low alpine zone in this region is generally classified as low shrub tundra (Walker et al., 2005) with mountain birch (Betula pubescens Ehrh.) forming the tree line (Oksanen & Virtanen, 1995). Topography affecting snow accumulation and moisture conditions creates habitats that are differentiated into exposed ridges, and steep and gentle parts of slopes, creating a sequence from xeric to mesic and very moist conditions with increasing duration of snow cover (Fremstad, 1997). These habitat characteristics give rise to distinct vegetation types such as ridge, mesic and snowbed vegetation (Fremstad, 1997). In this study we targeted mesic and snowbed habitats. Commonly occurring plant species in mesic habitats on the Varanger Peninsula include tall stature forbs (e.g., Alchemilla spp., Geranium sylvaticum L., Ranunculus acris L., Rhodiola rosea L.) in combination with grasses (e.g., Phleum alpinum L., Poa pratense ssp. alpigena (Fr.) Hiit., Festuca rubra L.). Snowbed habitats are characterized by prostrate Salix species (Salix herbacea L.) in combination with other grasses (e.g., Festuca rubra L., Poa alpina L.) and forbs (e.g., Cerastium sp.) of lower stature. Mosses such as Dicranum spp. or Polytrichum spp. are also prevalent here.

Semi-domesticated reindeer (Rangifer tarandus L.) that are managed by indigenous Sami people are the most common large herbivores in eastern Finnmark. In summer, reindeer herds are kept in the coastal mountains in large districts, which range in area from about 300 to 4000 km2, with most reindeer migrating inland during winter. Densities of reindeer have increased during the past two decades in some of these summer grazing districts, whilst remaining constant in others (see Table 2 in Ravolainen et al., 2010). This was evident on Varanger Peninsula during the period of our study, with contrasting reindeer densities observed in the two neighboring districts (Fig. 1). Other large herbivores present on Varanger peninsula are moose (Alces alces L.) and locally occurring domestic sheep (Ovis aries L.). Ptarmigans (Lagopus lagopus L. and Lagopus muta Montin), Norwegian lemming (Lemmus lemmus L.), root vole (Microtus oeconomus Pallas) and grey-sided vole (Myodes rufocanus Sund.) are also found in the area (Henden et al., 2011).

Sampling design

We employed a hierarchical, nested sampling design. Our protocol for selecting sampling units that corresponded to the habitats of interest involved several levels of selection (Fig. 1). Using the Varanger Peninsula as the sampling region (Fig. 1A) we covered both districts of contrasting reindeer density. We used information retrieved from a digital elevation model (DEM) to locate landscape areas that had potential sampling units representing the habitats of interest: Using GIS (ESRI ArcGIS with ArcMap. Version 8.3.0) we placed a raster of 2 × 2 km landscape areas over a 25 × 25 m pixel DEM (produced by Norwegian Mapping Authorities on the basis of elevation contour lines) covering the entire peninsula (Fig. 1A). Potential sampling units needed to have at least two 25 × 25 m neighboring pixels of concave topography with a mean slope between 5 °and 30 °. We restricted sampling to units that were a minimum distance of 500 m from birch forests and to an altitude of below 350 m above sea level in order to stay within the low alpine tundra. Finally we avoided lakes, glaciers, major roads and power lines, and only considered units that were within a one day’s walking distance from a road in order to be accessible. We then only selected landscape areas that according to the DEM included at least three potential sampling units that followed these criteria. This limited us to a total of 21 landscape areas over the whole peninsula. Out of time constraints we ultimately sampled nine of these landscape areas, divided between the two reindeer districts and with a good geographic spread (Fig. 1A).

Within each landscape area, the selection of sampling units was based on two different approaches of defining them (Fig. 1B). In the first approach (formal approach), we applied expert knowledge by defining a priori criteria in two steps. First, we defined topographical criteria to locate habitats in GIS (as described above). However, the spatial resolution of our DEM was too coarse for an a priori distinction of the two target habitats. Therefore, secondly, we defined additional criteria to be evaluated in the field. Here, the sampling unit had to show characteristics indicating both target habitats (i.e., mesic and snowbed) to be present. This criterion corresponded to a visible shift in plant species composition. In addition, the visually estimated vegetation cover had to be higher than 75%, and the habitat’s grain size had to be large enough to include a minimum of two transects for vegetation measurements (with at least one transect having a length of 10 m and every transect being 5 m apart; see more details below). If a potential sampling unit failed to meet any of these criteria, it was discarded and the next most accessible potential sampling unit was visited and inspected for possible field analyses. The sampling units of the formal approach correspond to the sampled habitats in González et al. (2010) and Ravolainen et al. (2010).

In the second approach (subjective approach), we based the selection of sampling units on a subjective definition as follows. As we entered the landscape areas, we subjectively assessed topography to locate sloping, concave terrain for the habitats of interest. When a typical plant community that either indicated a mesic or a snowbed habitat was found, it was considered as part of a sampling unit and it was analyzed as long as habitat size complied to the additional field criteria used in the formal approach (i.e., a vegetation cover of minimum 75% over a habitat area large enough to include a minimum of two transects, with at least one of them being 10 m long and each transect being horizontally spaced 5 m apart from each other). For both approaches, the final study unit was delineated either by convex areas of heath vegetation or a maximum transect length of 50 m.

Sometimes we sampled two sampling units per approach within one landscape area, in which case the closest set of sampling units, i.e., one from each of the two approaches, were termed “study area” being nested within landscape area (Fig. 1B).

Measurement of plant community characteristics

Within each selected habitat, measurement of plant community characteristics was identical for both approaches, except for the placement of transects. In the formal approach, the starting point of each transect was given by the initial GPS coordinates; in the subjective approach, starting points were chosen subjectively so that transects would cover the longest spatial extent of the targeted habitats (Fig. 1C). For both approaches, each transect was marked with a measuring tape running downslope from the starting point, with 5 m in horizontal distance between transects. Depending on the spatial extent of the habitats, we sampled between 2 and 5 transects with lengths varying from 4 m to 32 m. Thereafter, we recorded plant species abundance using the point intercept method according to Bråthen & Hagberg (2004). A frame of 40 cm × 40 cm with 5 pins of 2 mm diameter attached, one to each of the four frame corners and one to the center (see Ravolainen et al., 2010), was placed at fixed intervals of 2 m along the measuring tape. For each placement of the frame (i.e., for each plot), intercepts between pins and above ground vascular plant parts were recorded for each species separately. Species within the frame that were not hit by a pin were recorded with the value of 0.1. Table 1 presents a list of replication of all study units according to the spatial hierarchy of our design.

Table 1 The sample sizes are presented for each of the hierarchical levels of the sampling design, for each of the two approaches and their summarized sample size.

The formal and the subjective approach share samples at both levels above the level of sampling units.

	Nested hierarchy	Replication of units	
		Formal	Subjective	Total for both approaches	
Mesic habitat	Landscape area	9	9	9	
Study area	11	11	11	
Habitats/sampling units	11	11	22	
Transects	30	25	55	
Plots	199	152	351	
Snowbed habitat	Landscape area	6	6	6	
Study area	7	7	7	
Habitats/sampling units	7	7	14	
Transects	18	16	34	
Plots	85	103	188	

Response variables for data analyses

We converted point intercept data into biomass (g/plot) using weighted linear regression (Bråthen & Hagberg, 2004) and established calibration models (see Table S1 in Ravolainen et al., 2010), after which plant community measures were calculated for each plot in the data set. First we calculated three commonly used measures of within community (alpha-) diversity (Gini-Simpson index, Shannon entropy and species richness). Then we calculated biomass of the most dominant species (Betula nana L., Empetrum nigrum L. Hagerup. and Vaccinium myrtillus L.) and biomass of plant functional groups (as in Bråthen et al., 2007). Certain plant functional groups such as hemi-parasites had very low abundance and were therefore merged into the group of forbs (Table 2). Species and plant functional groups differed between the two habitats of interest, reflecting the fact that the mesic and the snowbed habitats were generally different in their species composition.

Table 2 Major plant functional groups and their associated species encountered in mesic and snowbed habitats.

The letters “M” (mesic) and “S” (snowbed) indicate the occurrence of each species within the respective target habitat. The nomenclature follows the Pan Arctic Flora (http://nhm2.uio.no/paf/).

Forbs		Grasses		
Alchemilla alpina (M,S)	Ranunculus acris (M, S)	Agrostis mertensii (M, S)	Juncus filiformis (M, S)	
Antennaria alpina (M)	Rhodiola rosea (M, S)	Anthoxanthum nipponicum (M, S)	Luzula multiflora (M, S)	
Antennaria dioica (M, S)	Rubus chamaemorus (M, S)	Avenella flexuosa (M, S)	Luzula spicata (M, S)	
Bartsia alpina (M, S)	Rumex acetosa (M, S)	Calamagrostis neglecta (M, S)	Luzula wahlenbergii (S)	
Bistorta vivipara (M, S)	Sagina saginoides (S)	Calamagrostis phragmitoides (M)		
Caltha palustris (M)	Saussurea alpina (M, S)	Festuca ovina (M, S)	Deciduous woody plants	
Chamaepericlymenum suecicum (M)	Saxifraga cespitosa (M)	Festuca rubra (M, S)	Arctous alpina (M)	
Campanula rotundifolia (M, S)	Sibbaldia procumbens (M, S)	Phleum alpinum (M, S)	Vaccinium uliginosum (M, S)	
Comarum palustre (M)	Silene acaulis (M)	Poa alpina (M, S)		
Draba glabella (M)	Solidago virgaurea (M, S)	Poa pratensis (M)	Evergreen woody plants	
Epilobium anagallidifolium (S)	Stellaria nemorum (S)	Vahlodea atropurpurea (M)	Andromeda polifolia (M)	
Epilobium hornemannii (M)	Taraxacum croceum aggregate (M, S)		Dryas octopetala (M)	
Euphrasia frigida (M,S)	Thalictrum alpinum (M, S)	Silica rich grasses	Harrimanellahypnoides (M, S)	
Euphrasia wettsteinii (M, S)	Trientalis europaea (M, S)	Deschampsia cespitosa (M, S)	Juniperus communis (M)	
Geranium sylvaticum (M, S)	Trollius europaeus (M, S)	Nardus stricta (M, S)	Kalmia procumbens (M, S)	
Geum rivale (M)	Veronica alpina (M, S)		Linnaea borealis (M)	
Listera cordata (M)	Viola biflora (M, S)	Sedges/Rushes	Orthilia secunda (M)	
Melampyrum sylvaticum (M)	Viola palustris (M)	Carex aquatilis (S)	Phyllodoce caerulea (M)	
Omalotheca norvegica (M, S)		Carex bigelowii (M, S)	Pyrola minor (M, S)	
Omalotheca supina (M, S)	Prostrate Salix species	Carex brunnescens (M)	Pyrola grandiflora (M, S)	
Oxyria digyna (S)	Salix herbacea (M, S)	Carex canescens (M, S)	Vaccinium vitis-idaea (M, S)	
Parnassia palustris (M, S)	Salix reticulata (M)	Carex lachenalii (M, S)		
Pedicularis lapponica (M, S)		Carex vaginata (M, S)	Dominant plant species	
Pinguicula vulgaris (M)		Eriophorum angustifolium (M)	Betula nana (M, S)	
Potentilla crantzii (M)		Eriophorum vaginatum (M)	Empetrum nigrum (M, S)	
Potentilla erecta (M)		Juncus arcticus (S)	Vaccinium myrtillus (M, S)	

Statistical analysis

We analyzed the three measures of (within-) community diversity and the biomass of different species and plant functional groups as response variables separately for each habitat type. When fitting linear mixed effect models, the approach to defining sampling units (formal versus subjective), the reindeer district (east versus west) and their interaction were used as fixed factors in the models. Bedrock type was included as a factor with three levels (sandstone; sandstone intermingled with schist; sandstone intermingled with schist and calcareous rock) and used as a co-variate (Table S2). The landscape areas and the study areas were set as random factors to account for spatial autocorrelation within areas. For some of the response variables we had to exclude study areas from the random effects structure because data existed for one study area per landscape area only. Models that had biomass of dominant plant species or biomass of functional groups as response variable were loge(x + v) transformed in order to assure model assumptions, with (v) representing the smallest biomass value of the sampled data in order to avoid negative values for plots with zero abundance. Diversity measures were not transformed. We used standard diagnostics to assess constancy and normality of residuals and controlled for outliers. All models were run using the lme function as part of the nlme package (Pinheiro et al., 2012) in R (version 2.12.1; R Development Team, 2010). A list of all models, containing Akaike’s Information Criterion and test statistics for the used fixed factors, can be found in Tables S3 and S4.

Results

Mesic habitat

The approach to defining sampling units affected almost all estimates of plant community diversity in the mesic habitat (Figs. 2A–2C). The estimates of the diversity indices were in most cases significantly higher in the subjective compared with the formal approach. However, for one of the indices (Gini-Simpson), estimates were only higher in the western district (Fig. 2A).

Figure 2 The figure represents all model estimates for the mesic habitat.

Effect sizes (mean ±95% confidence interval) of the response difference between the subjective and the formal approach of defining sampling units within the mesic habitat are shown for estimates of diversity (A, B, C) and estimates of biomass of dominant plant species and functional groups (D). Effect sizes above or below the dotted line can be interpreted as the subjective approach having higher or lower estimates than the formal approach. Effect sizes of biomass estimates are back transformed values from a logarithmic scale, using the exponential on effect sizes from our model, and may be interpreted as the ratio of the subjective/formal approach. The numbers at the base of each figure represent estimates of the respective diversity index (A, B, C) and the geometric mean of the biomass estimates (D) from the formal approach for each respective response variable. Geometric means can be interpreted as approximate biomass estimates for the respective district.

Estimates of plant functional group biomass and biomass of dominant plant species were significantly different between the two approaches (Fig. 2D). The biomass of forbs was estimated to be consistently higher when using the subjective approach in both districts. However, there were interaction effects between the approach type and the reindeer district. For many response variables, differences between the two approaches were only significant in one of the two districts (prostrate Salix, grasses, evergreens, deciduous woody species, Vaccinium myrtillus, Empetrum nigrum L.). Biomass estimates of other response variables (silica rich grasses and Betula nana) were lower in the eastern, but higher in the western district when the subjective approach was used.

Snowbed habitat

The approach to defining sampling units also had significant effects on the diversity estimates for the snowbed habitat (Figs. 3A–3C). For both Shannon entropy and species richness, the subjective approach revealed higher estimates in the eastern but lower estimates in the western district (Figs. 3B and 3C).

Figure 3 The figure represents all model estimates for the snowbed habitat.

Effect sizes (mean ± 95% confidence interval) of the response difference between the subjective and the formal approach of defining sampling units within the snowbed habitat are shown for estimates of diversity (A, B, C) and estimates of biomass of dominant plant species and functional groups (D). Effect sizes above or below the dotted line can be interpreted as the subjective approach having higher or lower estimates than the formal approach. Effect sizes of biomass estimates are back transformed values from a logarithmic scale, using the exponential on effect sizes from our model, and may be interpreted as the ratio of the subjective/formal approach. The numbers at the base of each figure represent estimates of the respective diversity index (A, B, C) and the geometric mean of the biomass estimates (D) from the formal approach for each respective response variable. Geometric means can be interpreted as approximate biomass estimates for the respective district, hence the slightly negative value for Empetrum nigrum which had very low biomass recordings in the eastern district.

Significant differences between the two approaches were also found for the biomass estimates of dominant plant species and of different plant functional groups (Fig. 3D). Similar to the mesic habitat, there were significant interaction effects between the approach to define sampling units and the reindeer district. Biomass estimates of some plant functional groups were only affected by the approach in one of the two districts (forbs, grasses, silica rich grasses). For prostrate Salix, we found opposite effects of the approach between the two districts. The biomass was estimated to be significantly lower in the eastern, but significantly higher in the western district when using the subjective approach.

Discussion

Differences in defining sampling units affect community estimates depending on ecological context

In our study, the sampling approach based on a subjective definition of sampling units revealed significant effects on many of our response variables in comparison to the approach based on formal rules.

For instance, from our subjective approach our conclusion would be that mesic and snowbed habitats had very low but comparable biomass of silica rich grasses within the two reindeer districts where data were collected. In contrast, our results based on a formal definition of sampling units show a considerably higher abundance of silica rich grasses in the eastern district where also reindeer density is higher. The role of silicate rich plants in plant herbivore interactions (Vicari & Bazely, 1993) indicate that the acceptance of one conclusion or the other could lead to very different ecological outcomes and highlight the need for careful consideration in the definition of sampling units in ecological studies. Hence, the way sampling units were defined begets ecological conclusions to be drawn (Peterson et al., 2001).

Previous studies have documented how individual preferences for certain sampling units could result in biased estimates, with for instance higher estimates of species richness compared to probabilistic sampling approaches (Chytrý, 2001; Botta-Dukát et al., 2007; Diekmann, Kühne & Isermann, 2007). However, the subjective selection in this study only rendered constantly higher estimates of species richness in the mesic habitats, while species richness in the snowbed habitats was only increased by the subjective approach in the eastern district. We can only speculate on the reasons for this lack of consistency. For the mesic habitat, the consistently higher estimates of species richness in the subjective approach might be due to the fact that we focused on habitats with many indicator species that can be easily distinguished visually, such as different forb species (see Fig. 2D). Such a preference could also explain the higher estimates of species richness and forbs of snowbeds in the eastern district, where high reindeer abundance might lead to generally low abundance of facilitating plant species such as forbs (Bråthen et al., 2007). The lower species richness estimates of the snowbed habitat in the western district might be due to a preference of the sampling units that were visually more strongly impacted by snow, causing a higher probability of selecting for late snowbeds as opposed to earlier snowbeds. Late emergence from snow causes marginal growing conditions for vascular plants and reduced species richness (Björk & Molau, 2007). However, the fact that these interpretations would only account for one specific district shows that the bias caused by the subjective definition of sampling units in species richness depends on ecological context. We found similar context dependencies for other diversity indices and for many of the biomass response variables in our study (Figs. 2 and 3).

How to define sampling units to ensure comparability between studies?

Context dependency of the differences in estimates between the two approaches could also have relevance to the comparability of ecological studies. Idiosyncratic results from work on similar study systems are often found in ecological research (Chase et al., 2000; Hedlund et al., 2003; Badano & Cavieres, 2006). Our results indicate that idiosyncratic results within studies or among different studies may have their roots in the way sampling units have been defined. With context dependency being one of the greatest challenges of ecology today (Wardle et al., 2011), additional context dependency enforced by the way ecological sampling units are defined will make it even more difficult to tackle this challenge (see e.g., Franklin, Noon & George, 2002).

The definition of sampling units in our formal approach involved abiotic characteristics known to represent the habitats in question (e.g., slope and curvature). Such terrain criteria were applied in a way that allowed us to accurately document each sampling unit characteristic, although at the coarse scale of the DEM. In contrast, we did not apply biotic criteria such as the usage of indicator plant species or indicator functional groups in an a priori way in this approach, for two reasons. First, plant composition was largely unknown across the potential sampling units of the two habitats, reflecting the absence of vegetation maps (at the grain size of our habitats) for the study area. Secondly, any preference for plant indicators was likely to interfere with the outcome of our research question (Ewald, 2003), i.e., what are the plant community characteristics of mesic and snowbed habitats? However, because our focus was on plants, simple biotic criteria of vegetation cover and a visual shift in type of plant community were not considered to interfere with our conclusions. Although the rules applied in the formal approach were quite simple, they were considered relevant to the research questions set. Clearly, more specific research questions would demand more refined formal rules.

For applications in ecology, the reproducibility of studies and the comparison between studies are essential (Shrader-Frechette & McCoy, 1994). Therefore, for any true comparison between studies to be made, discrete sampling units such as habitats must be defined in the same way (Loehle, 2004). Our study shows that even slight deviations in the definition of sampling units could affect the comparability of results, even within the same study system. That is, only the formal approach to defining sampling units is concomitantly transparent (i.e., by the set of formal rules applied), and produced results that fulfill the premise on which further ecological understanding can be developed. Hence, as sampling procedures that allow reproducibility and comparisons between studies are essential, so are the sampling procedures to allow accumulation of ecological knowledge. Therefore, we believe that the call for formal approaches in study designs (Legendre et al., 2002; Edwards et al., 2005; Edwards et al., 2006; Albert et al., 2010) should also be extended to formal approaches to the definition of sampling units.

The application of expert knowledge is a matter of discussion in several fields of ecology. There are a number of studies that address ways of eliciting expert knowledge for decision making in conservation or landscape ecology (Burgman et al., 2011; Martin et al., 2011; McBride & Burgman, 2012), including the use of expert opinion for modeling (Booker & McNamara, 2004; Kuhnert, Martin & Griffiths, 2010; Martin et al., 2011). In landscape ecology, the use of expert knowledge has recently been challenged to adhere to the same scientific rigor as other sampling approaches (Morgan, 2014). We believe the application of expert knowledge deserves equal attention in terms of the definition of sampling units, and especially in the definition of habitats, which should be done in a transparent way (Whittaker, Levin & Root, 1973; Franklin, Noon & George, 2002).

Supplemental Information

Data S1 The dataset that was used for the analyses

Click here for additional data file.

Table S1 Number of habitats per district and approach and their corresponding bedrock type

We used information obtained from bedrock maps (The Geological survey of Norway; www.ngu.no), assigning each target habitat with the correct bedrock type after the field session.

Click here for additional data file.

Table S2 Replication of plots according to fixed factors of the mixed models

Mixed models contained the approach of defining sampling units, the two districts of different reindeer density and the three bedrock categories as fixed factors.

Click here for additional data file.

Table S3 Linear Mixed Effect Models for the mesic habitat type

The table shows Akaike’s Information Criterion (AIC) for each model. “Value” indicates effects of factor levels compared to the Intercept which is followed by a t-test statistic.

Click here for additional data file.

Table S4 Linear Mixed Effect Models for the snowbed habitat type

The table shows Akaike’s Information Criterion (AIC) for each model. “Value” indicates effects of factor levels compared to the Intercept which is followed by a t-test statistic.

Click here for additional data file.

We are grateful to Geir Vie for his assistance during data collection in the field and to COAT (Climate-Ecological Observatory of Arctic Tundra) for inspiration.

Additional Information and Declarations

Competing Interests

Author Contributions

Virve T. Ravolainen is an employee at the Norwegian Polar Institute and Leif Einar Støvern is an employee at the Norwegian Institute for Forest and Landscape Research. Nigel G. Yoccoz is an Academic Editor for PeerJ.

Martin A. Mörsdorf analyzed the data, wrote the paper, prepared figures and/or tables, reviewed drafts of the paper.

Virve T. Ravolainen and Kari Anne Bråthen conceived and designed the experiments, performed the experiments, analyzed the data, wrote the paper, reviewed drafts of the paper.

Leif Einar Støvern conceived and designed the experiments, performed the experiments, wrote the paper, prepared figures and/or tables, reviewed drafts of the paper.

Nigel G. Yoccoz analyzed the data, wrote the paper, reviewed drafts of the paper.

Ingibjörg Svala Jónsdóttir wrote the paper, reviewed drafts of the paper.

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
