# Peer review of "Definition of sampling units begets conclusions in ecology: the case of habitats for plant communities"

_PeerJ, doi:10.7717/peerj.815_

## Round 0.1 · original submission · Minor Revisions

Your paper only needs minor corrections, please follow the suggestions made by both reviewers. Improving the methods section was specifically suggested by Rev. 2 .

·

Basic reporting

I believe the submission is of high quality relative to PeerJ's standards of basic reporting. The English is excellent, and I have annotated the MS (attached) with notes that the authors can use to correct minor grammatical errors or areas that require clarification.

My principal concern is with the proliferation of nomenclature in the MS, especially with respect to terms containing the word "expert." As I began to read through the introduction, I found it a bit confusing to separate three terms: expert knowledge, formal definition of sampling units, and subjective definition of sampling unit. I think the authors need to make it clear that “expert knowledge” is an umbrella term applied to both formal and subjective sampling approaches. It took me a while to understand this simple nomenclatural concept. On line 40, the concept is made less clear by introduction of a new term, “expert opinion.” Later, the introduction of new terms, “knowledge” (with no modifier) and “expert judgment,” in the paragraph beginning on line 62 further “muddies the waters.”

Experimental design

I believe the experimental design is rigorous and meets PeerJ's standards.

Validity of the findings

While the results are certainly valid, I think the authors should do better job of stating why their work advances an area where some previous work exists. In the sentence beginning on line 44, the authors cite previous studies (Chytrý, 2001; Botta-Dukát et al., 2007; Hédl, 2007) that demonstrate biased results from subjective sampling. I would thus like to see some statement of what this research adds to the subject beyond what has already been discovered by previous researchers.

Additional comments

In the sentence beginning on line 91, how can an average vary? Should there not be just a single value reported for average precipitation?

I don't think the raster of 2x2 km landscape areas will be clear at the size of reproduction of Fig. 1.

I do not fully understand the statement beginning on line 135. Is the study blocked by bedrock type? If so, there would be three blocks in each reindeer district, rather than 4 in one and 5 in the other.

The sentence beginning on line 169 needs attention for clarity’s sake. In particular, what is a ribbon?

Reviewer 2 ·

Basic reporting

No comments.

Experimental design

How was the digital elevation model constructed, which are the source data of the DEM? As you state later on, the habitats occur on a smaller spatial scale than represented by the DEM. For repeatability, an idea of the "quality" of the DEM is interesting.
The formal selection of sampling units is not clearly explained
- were the topographical criteria for locating mesic and snowbed habitats the same? (i.e. anything between 5 and 30 deg.)? In the discussion you state that "the definition of habitats in our formal approach involved abiotic characteristics known to represent the habitats in question (slope, curvature, altitude)". The predefined definitions of these criteria, in particular relating to slope, is however missing.
- "the sampling unit had to show characteristics indicating both target habitats" - what does this mean? Are the two habitats sampled simultaneously - do you find both habitats within a 25 x 25 m pixel? Isn't the sampling unit one habitat?
- "this criterion corresponded to a visible shift in plant communities" - for repeatability of the study, this should be expressed in a better way. How do you actually delineate the sampling unit in space?
The subjective selection procedure:
- did the 75 % vegetation cover criterion also apply here?
Statistical analyses
- do you treat the plots within a sampling unit as independent, or do you take into account a spatial structure/autocorrelation following the placement of plots along transects?

Validity of the findings

No comments.

Additional comments

This is a well-written and well-founded article, the presentation of results is good and the discussion of the results well-linked to the stated research questions.

Minor comments:
lines 103-107. sp. and ssp. should not be in italics
line 151: better than "the same habitats as in" is "the sampled habitats in"
line 236: you found opposite effects among/between the two districts, not within
line 301-302: replace it are with it is, or rephrase the sentence.

---

## Round 0.2 · accepted · Accept

Your paper is ready to be published, all suggestions have been taken.